# Nanoantennas Inversely Designed to Couple Free Space and a Metal–Insulator–Metal Waveguide

**DOI:** 10.3390/nano11123219

**Published:** 2021-11-26

**Authors:** Yeming Han, Yu Lin, Wei Ma, Jan G. Korvink, Huigao Duan, Yongbo Deng

**Affiliations:** 1State Key Laboratory of Applied Optics (SKLAO), Changchun Institute of Optics, Fine Mechanics and Physics (CIOMP), Chinese Academy of Sciences, Changchun 130033, China; hanyeming@ciomp.ac.cn (Y.H.); linyu@ciomp.ac.cn (Y.L.); 2State Key Laboratory of Modern Optical Instrumentation, College of Optical Science and Engineering, Zhejiang University, Hangzhou 310027, China; 3Institute of Microstructure Technology (IMT), Karlsruhe Institute of Technology, 76344 Karlsruhe, Germany; jan.korvink@kit.edu; 4National Engineering Research Center for High Efficiency Grinding, State Key Laboratory of Advanced Design and Manufacturing for Vehicle Body, College of Mechanical and Vehicle Engineering, Hunan University, Changsha 410082, China; duanhuigao@163.com

**Keywords:** nanoantenna, topology optimization, metal–insulator–metal waveguide, surface plasmon

## Abstract

The metal–insulator–metal (MIM) waveguide, which can directly couple free space photons, acts as an important interface between conventional optics and subwavelength photoelectrons. The reason for the difficulty of this optical coupling is the mismatch between the large wave vector of the MIM plasmon mode and photons. With the increase in the wave vector, there is an increase in the field and Ohmic losses of the metal layer, and the strength of the MIM mode decreases accordingly. To solve those problems, this paper reports on inversely designed nanoantennas that can couple the free space and MIM waveguide and efficiently excite the MIM plasmon modes at multiple wavelengths and under oblique angles. This was achieved by implementing an inverse design procedure using a topology optimization approach. Simulation analysis shows that the coupling efficiency is enhanced 9.47-fold by the nanoantenna at the incident wavelength of 1338 nm. The topology optimization problem of the nanoantennas was analyzed by using a continuous adjoint method. The nanoantennas can be inversely designed with decreased dependence on the wavelength and oblique angle of the incident waves. A nanostructured interface on the subwavelength scale can be configured in order to control the refraction of a photonic wave, where the periodic unit of the interface is composed of two inversely designed nanoantennas that are decoupled and connected by an MIM waveguide.

## 1. Introduction

Plasmonic devices have been intensely studied in the recent years from both fundamental and applied perspectives [1,2,3,4]. The potential of the plasmon waveguide to guide subwavelength optical modes at metal–dielectric interfaces, known as so-called surface plasmon polaritons, has been shown. Several different plasmonic waveguiding structures have been proposed [5,6,7,8,9,10]. The highly confined mode that is only present near the surface plasma frequency is the focus of most of these structures. In this regime, the optical mode typically has low group velocity and short propagation length. However, it has been shown that an MIM structure with a dielectric regional thickness of ~100 nm can support a propagating mode with a nanoscale modal size at a wavelength range extending from deep centimeter (DC) to visible [11]. Therefore, such a waveguide acts as an important interface between conventional optics and subwavelength photoelectrons; particularly, it has attracted attention for applications ranging from subwavelength optical routing and control [12,13,14] to molecular spectroscopy [15] and even as a negative index waveguide [16].

The strong field confinement and significant field enhancement that are provided by plasmon modes have attracted interest in MIM waveguides. The dispersion relations for the propagating plasmon modes in MIM structures have been studied analytically and numerically [11,17,18]. While theoretical studies of coupling to MIM plasmon waveguides using endface dielectric waveguides have been reported [19], the possibility of effectively free space coupling directly to the MIM plasmon mode has been overlooked, except in the case of subwavelength diffraction grating-based coupling [20], as shown in Appendix A, Figure A1. Sharp decreases in reflectance indicate that the conversion of photons to plasmons, which nonradiatively decay, has taken place; the resulting large wave vector mismatch between the MIM plasmon mode and photons makes optical coupling difficult. Further, the MIM mode strength decreases as the wave vector increases, this is due to more of the field residing in the metal layers, increasing Ohmic losses. In order to address this problem, plasmonic nanoantennas are an ideal selection as they can realize the coupling of the free space and MIM waveguide, which can efficiently focus a wide spectrum in order to form ultrafast hotspots on a nanometer scale [21,22].

Plasmonic nanoantennas require complex geometric and spatial arrangement; a structural design that is based on physical methods alone is not sufficient. Therefore, we have to rely on another approach: electromagnetic modeling based on a numerical simulation method. Among numerical simulation methods, reverse design methods aim to directly retrieve the appropriate structure in order to obtain the desired optical properties, so that the non-intuitive, irregular shape of the objective structure can be found. This is better than the empirical design of the structure in many applications [23]. Researchers have developed a number of reverse design methods for designing nanoantennas, such as the Multi-Objective Lazy Ant Colony Optimization (MOLACO) approach (this method was used to inversely design the nanoantenna in the present study), which achieves high transmission performance and a wide range for phase selection [24]; and the genetic algorithm (GA) approach (this method was used to inversely design the multi-slot energy harvesting nanoantenna), which can enhance the electric field in order to increase the absorptivity of the nanoantenna [25]. However, these methods have a relatively strong dependence on the selection of the initial population and they cost more CPU-time when a large number of individuals are involved. Therefore, this paper reports on inversely designed nanoantennas that can couple the free space and MIM waveguide while also efficiently exciting the MIM plasmon modes at multiple wavelengths and under the oblique incidence, by implementing an inverse design procedure using the topology optimization approach, as shown in Figure 1a. A 2D model of the MIM waveguide was considered, as shown in Figure 1b. The model consists of a domain of free space $\Omega_{f}$, the sandwiched MIM waveguide that is composed of two Au domains $\Omega_{m}$ and a free space domain $\Omega_{w}$, and a design domain $\Omega_{d}$ at the left endface of the MIM waveguide. The design domain $\Omega_{d}$ is used to inversely design the nanoantenna by using a topology optimization method and the metal material of the nanoantenna is Au. The light is coupled from the free space $\Omega_{f}$ to the inversely designed nanoantenna. The match between the MIM plasmon mode and photons in the free space is changed by a distribution of Au within the design domain. The results of the simulation are shown in Appendix B, Figure A2, Figure A3, Figure A4, Figure A5, Figure A6 and Figure A7.

Topology optimization is a computational tool that was originally developed for mechanical design problems, it has the advantages of clear logic, simple programming process and easy integration with the existing commercial finite element software [28,29]. In addition to mechanics, the method is very versatile and has been applied to a wide range of areas such as micro-electro-mechanical systems [30], acoustics [31,32,33], nano-photonics [34], and plasmonics [35,36,37]. Topology optimization works by varying the spatial distributions of different materials within a bounded design domain. The design problem is formulated as an optimization problem with the goal of finding the material’s distribution, minimizing or maximizing an objective function and measuring the quantity to be optimized. The design is described by a pixel or voxel representation in which a design variable is associated with each mesh element. Consequently, no explicit design parameterization is needed. The design is changed without geometrical constraints, such as those that are imposed when using a parameter-based shape or size optimization method.

The remainder of this paper is organized as follows. In Section 2, the topology optimization method is described for the inverse design of nanoantennas localized at the endface of an MIM waveguide; the results obtained using this method for the various nanoantennas are presented and discussed in Section 3; and, finally, the paper is concluded in Section 4.

## 2. Topology Optimization Method

The 2D model of the MIM waveguide, as shown in Figure 1b, was excited by an $H_{z}$-polarized plane wave propagating in the positive x-direction with incident angle θ. Assuming translational symmetry in the z-direction and linear polarization of the incident magnetic field, the total magnetic field was derived by solving the scalar Helmholtz equation using the finite element method [38]. All the materials were assumed to be non-magnetic and the relative magnetic permeability was set to $\mu_{r}=1$. The complex relative electric permittivity of Au ($\epsilon_{r}^{Au}$) was calculated by using the refractive index η and extinction coefficient κ, as $\epsilon_{r}^{Au}=\eta^{2}-\kappa^{2}+2j\eta \kappa$, where η and κ are taken from Johnson [39]. The time-harmonic factor $e^{j\omega t}$ is the imaginary unit, $j=\sqrt{-1}$ is the angular frequency and ω is the time.

In this 2D model, the $H_{z}$-polarized plane wave was described by the wave equation [40,41]:(1)$$\nabla \cdot \left[\epsilon_{r}^{-1}\nabla \left(H_{sz}+H_{iz}\right)\right]+k_{0}^{2}\mu_{r}\left(H_{sz}+H_{iz}\right)=0,\;\mathrm{in}\;\Omega$$
where ∇ is the gradient operator in the Cartesian coordinate system; $H_{z}=H_{sz}+H_{iz}$ is the total field, $H_{sz}$ and $H_{iz}$ are the scattering and incident fields, respectively; $\epsilon_{r}$ is the relative permittivity; $k_{0}=\omega \sqrt{\epsilon_{0}\mu_{0}}$ is the free space wave number in which $\epsilon_{0}$ and $\mu_{0}$ represent the free space permittivity and permeability, respectively; Ω is the domain that is enclosed by the perfectly matched layers (PMLs, as shown in the Appendix A, Figure A1). The infinite computational space was truncated by the PMLs, which were implemented by solving the wave equations by using the complex-valued coordinate transformation [38,42,43],
(2)$$\nabla_{x^{\prime }}\cdot \left(\epsilon_{r}^{-1}\nabla_{x^{\prime }}H_{sz}\right)+k_{0}^{2}\mu_{r}H_{sz}=0,{\;\mathrm{in}\;\Omega }_{P}\;$$
where $x^{\prime }$ is the complex-valued coordinate that has been transformed from the original Cartesian coordinate in $\Omega_{P}$; $\nabla_{x^{\prime }}$ is the gradient operator in the PMLs with transformed coordinates; and $\Omega_{P}$ is the union of the PMLs sketched in the 2D model of MIM waveguide. The transformed coordinates and the original Cartesian coordinates satisfied the transformation
(3)$$x^{\prime }=Tx,\forall x\in \Omega_{P}$$
with the transformation matrix [38]
(4)$$T=\left\{\begin{array}{l} \left(\begin{array}{cc} \left(1-j\right)\lambda /d & 0\\ 0 & 1 \end{array}\right),{\;\mathrm{in}\;\Omega }_{l}\cup \Omega_{r}\\ \left(\begin{array}{cc} 1 & 0\\ 0 & \left(1-j\right)\lambda /d \end{array}\right),{\;\mathrm{in}\;\Omega }_{t}\cup \Omega_{b}\\ \left(\begin{array}{cc} \left(1-j\right)\lambda /d & 0\\ 0 & \left(1-j\right)\lambda /d \end{array}\right),{\;\mathrm{in}\;\Omega }_{c} \end{array}\right.$$
where x is the original Cartesian coordinate; λ is the incident wavelength in the truncated background; and d is the thickness of the PMLs. The no-jump boundary condition for the scattering field is imposed on the interface between $\Omega_{P}$ and Ω
(5)$$\epsilon_{r}^{-1}\left(\nabla H_{sz}-\nabla_{x^{\prime }}H_{sz}\right)\cdot n=0,\;\mathrm{on}\;\partial \Omega$$
where n is the outward unitary normal on the domain boundary. The perfect magnetic ($H_{sz}=0$) condition was imposed on the external boundaries $\Gamma_{D}=\partial \left(\Omega \cup \Omega_{P}\right)$ of the PMLs.

Because the desired result of inversely designing the nanoantenna was to couple the free space and MIM waveguide and efficiently transfer the photons into MIM plasmon modes for multi wavelengths or multi oblique angles, the objective function for topology optimization was set to maximize the minimal value of the effective apertures corresponding to the specified wavelengths or oblique angles. This was equivalent to maximizing the corresponding minima of the normalized averaged energy flux density in the MIM waveguide
(6)$$\Phi =\underset{\lambda \in \left\{\lambda_{1},\lambda_{2}\cdots \lambda_{N_{1}}\right\}}{min}\frac{1}{\Phi_{0}\left(\lambda \right)}\frac{\int_{{Ω}_{w}}S\left(\lambda \right)\cdot e_{x}dΩ}{L_{W}}$$
for the multi wavelengths nanoantenna and
(7)$$\Phi =\underset{\theta \in \left\{\theta_{1},\theta_{2}\cdots \theta_{N_{2}}\right\}}{min}\frac{1}{\Phi_{0\left(\theta \right)}}\frac{\int_{{Ω}_{w}}S\left(\theta \right)\cdot e_{x}dΩ}{L_{W}}$$
for the multi oblique angles nanoantenna, where $\left\{\lambda_{1},\lambda_{2}\cdots \lambda_{N_{1}}\right\}$ is the admittable set of the incident wavelengths; $\left\{\theta_{1},\theta_{2}\cdots \theta_{N_{2}}\right\}$ is the admittable set of the incident angles; $L_{w}$ is the length of the considered MIM waveguide; S is the averaged Poynting vector $\frac{1}{2}Re\left(E\times H^{*}\right)$ with Re and * representing the operators used to extract the real part and implement the conjugate of a complex, respectively; $e_{x}$ is the direction vector of the x-axis; and $\Phi_{0}$ is the normalizer, the averaged energy flux density corresponding to the direct coupling of the free space and MIM waveguide.

In topology optimization, the density approach [28] relaxes the discrete nature of placing different materials within individual mesh elements by introducing a set of elementwise-constant material density $\gamma_{p}$, which is allowed to take any continuous value between 0 and 1. This material density can be derived from subsequent filtering, elementwise and projection, of a design variable γ that is defined on the design domain [44]. The filtering is implemented by the partial differential equation (PDE) filter
(8)$$\begin{array}{l} -\nabla \cdot \left(r^{2}\nabla \gamma_{f}\right)+\gamma_{f}=\gamma ,{\;\mathrm{in}\;\Omega }_{d}\\ n\cdot \nabla \gamma_{f}=0,\;\mathrm{in}\;\partial \Omega_{d} \end{array}$$
where r is the filtering radius and $\gamma_{f}$ is the filtered design variable. The elementwise operation, while deriving the elementwise design variable $\gamma_{e}$, is carried out by averaging the filtered design variables for every piece of the design domain, which is divided into discrete pieces [44]
(9)$$\begin{array}{l} y_{e}=\sum_{n=1}^{N}\frac{1}{V_{n}}\int_{P_{n}}y_{f}dΩ\\ \mathrm{with}V_{n}=\int_{P_{n}}1dΩ,\cup_{n=1}^{N}P_{n}={Ω}_{d}\\ \mathrm{and}P_{l}\cup P_{m}=\emptyset \left(l,m=1,2\cdots N,l\neq m\right) \end{array}$$
the projection of the elementwise design variable is implemented by the threshold method [45,46]
(10)$$\gamma_{p}=\frac{\tan h\left(\beta \xi \right)+\tan h\left(\beta \left(\gamma_{e}-\xi \right)\right)}{\tan h\left(\beta \xi \right)+\tan h\left(\beta \left(1-\xi \right)\right)}$$
where $\xi \in \left[0,1\right]$ and β are the threshold and projection parameters, respectively; for the choices of the values of ξ and β, one can refer to [45,46]. During the evolutionary process of the design variable, the material density was forced to converge towards a 0−1 distribution, which represented the material distribution [47]; and the relative electric permittivity inside the design domain was interpolated using the hybrid of the logarithmic and power law approaches [37]
(11)$$\epsilon_{r}\left(\gamma_{p}\right)=10^{\log\epsilon_{r}^{Au}-\frac{1-\gamma_{p}^{3}}{1+\gamma_{p}^{3}}\left(\log\epsilon_{r}^{Au}-\log\epsilon_{r}^{f}\right)},{\;\mathrm{in}\;\Omega }_{d}$$
with $\gamma_{p}=0$ and $\gamma_{p}=1$ representing Au ($\epsilon_{r}^{Au}$) and free space ($\epsilon_{r}^{f}$), respectively. This material interpolation can mimic the exponential decay of the electromagnetic field at the Au-free space interface.

Topology optimization is a gradient-based method and relies on adjoint sensitivity analysis for the efficient calculation of the design sensitivities [28]. Based on the adjoint method for the partial differential equation constrained optimization problem [48], the adjoint sensitivity for the design objectives in Equations (6) and (7) can be derived to be
(12)$$\delta \Phi =-\int_{\Omega_{d}}{\hat{y}}_{f}\delta ydΩ,y\in L^{2}\left({Ω}_{d}\right)$$
where δγ is the first order variational of γ; ${\hat{\gamma }}_{f}$ is adjoint variable of the filtered design variable $\gamma_{f}$; and $L^{2}\left(\Omega_{d}\right)$ is the second order Lebesque space for the real functions defined on $\Omega_{d}$. ${\hat{\gamma }}_{f}$ is derived by sequentially solving the adjoint equations: find ${\hat{H}}_{sz}$ with ReH^sz∈HΩ∪ΩP,ImH^sz∈HΩ∪ΩP and ${\hat{H}}_{sz}=0$ on $\Gamma_{D}$, satisfying
(13)$$\begin{array}{l} \int_{\Omega }\frac{1}{\Phi_{0}L_{w}}[\left(\frac{\partial \left(S\cdot e_{x}\right)}{\partial Re\left(H_{SZ}\right)}-j\frac{\partial \left(S\cdot e_{x}\right)}{\partial Im\left(H_{SZ}\right)}\right)\Phi \\ \left.+\left(\frac{\partial \left(S\cdot e_{x}\right)}{\partial \nabla Re\left(H_{sz}\right)}-j\frac{\partial \left(S\cdot e_{x}\right)}{\partial \nabla Im\left(H_{sz}\right)}\right)\cdot \nabla \phi \right]\\ -\epsilon_{r}^{-1}\nabla {\hat{H}}_{sz}^{*}\cdot \nabla \phi +k_{0}^{2}\mu_{r}{\hat{H}}_{sz}^{*}\phi d\Omega \\ {+\int_{\Omega_{P}}-\epsilon_{r}^{-1}\left(T\nabla {\hat{H}}_{sz}^{*}\right)\cdot (T\nabla \Phi )|T|}^{-1}\\ +k_{0}^{2}\mu_{r}{\hat{H}}_{sz}^{*}\Phi |T|d\Omega =0,\forall \Phi \in H\left(\Omega \cup \Omega_{P}\right) \end{array}$$

Find ${\hat{\gamma }}_{f}\in H\left(\Omega_{d}\right)$, satisfying (14)∫Ωdr2∇y^f·∇φ+y^fφ−[∑n=1N1Vn∫PnRe(∂ϵr−1∂yp∂yp∂ye∇(Hsz+Hiz))·Re ∇H^sz*−Im∂ϵr−1∂γp∂γp∂γe∇Hsz+Hiz·Im∇H^sz*dΩφdΩ=0,∀φ∈HΩd where ${\hat{H}}_{sz}$ is the adjoint variable of $H_{sz}$; $\Gamma_{D}$ is the perfect magnetic conductor boundary $\partial \left(\Omega \cup \Omega_{P}\right)$; $H\left(\Omega \cup \Omega_{P}\right)$ and $H\left(\Omega_{d}\right)$ are the first order Hilbert spaces for the real functions defined on $\Omega \cup \Omega_{P}$ and $\Omega_{d}$, respectively; and Im is the operator used to extract the imaginary part of a complex.

Based on the derived adjoint sensitivity, a MATLAB implementation was used to solve the discretized model problem, using the finite element method of COMSOL Multiphysics (version 5.6), and the optimization was performed using the method of moving asymptotes (MMA) [49]. Finite element solving was implemented using 2 nm linear elements to resolve the surface plasmons that were associated with the dielectric–metal interfaces. The procedure for the evolutionary convergence of the topology optimization problems included the following steps: (a) solving the wave equations with the current design variable, for all of the incident wavelengths or oblique angles in the corresponding admittable set; (b) computing the normalized averaged energy flux densities in the MIM waveguide and setting the minima of the normalized averaged energy flux densities to be the objective value; (c) solving the adjoint equations based on the solution of the wave equations corresponding to the objective value; (d) computing the adjoint sensitivity of the design objective; (e) updating the design variable using the MMA; (f) if the stopping criterion is not satisfied, the loop returns to (a). In this solution procedure, the filter radius r of the PDE filter in Equation (8) was set to 20 nm; the threshold parameter ξ in Equation (10) was set to 0.5; and the initial value of the projection parameter β was set to 1 and doubled after every 20 iterations until the preset maximal value of $2^{8}$ is reached. The stopping criterion was specified to be the change of the objective values in five consecutive iterations satisfying $\frac{1}{5}\overset{4}{\underset{i=0}{\sum }}\left|\Phi_{k-i}-\Phi_{k-i-1}\right|/\left|\Phi_{k}\right|\leq 1\times 10^{-3}$ and $\beta \geq 2^{8}$ in the k-th iteration, where $\Phi_{k}$ is the objective value computed in the k-th iteration. For more details on implementing this topology optimization framework, the reader is referred to the references [50].

## 3. Results and Discussion

An inversely designed nanoantenna for a single frequency has a strong dependence on and sensitivity to the incident wavelength and oblique angle. Its spectral peak presents at the position corresponding to those two parameters. To decrease this dependence on and sensitivity to the incident wavelength, the coupling nanoantennas can be inversely designed by maximizing the minimal value of the normalized averaged energy flux density in the MIM waveguide with the incident wavelength in a specified range. Figure 2a shows the derived nanoantenna for the incident waves with a wavelength in the range from 850 nm to 1550 nm and the distribution of the real part of the magnetic field that is plotted for the four communication wavelengths. The spectra of the derived nanoantenna are plotted in Figure 2b as the function of the wavelength and oblique angle. From these spectra, we can confirm the decreased dependence on the incident wavelength in the specified range. The emitting performances of the derived nanoantenna are presented by the polar diagrams in Figure 2c. Figure 2c shows that the inversely designed nanoantenna can be used for several different incident waves while still keeping its directivity and the enhancement of the coupling efficiency.

To decrease the level of dependence on and sensitivity to the oblique angle, the coupling nanoantennas can be inversely designed by maximizing the minimal value of the normalized, averaged energy flux density in the MIM waveguide with an oblique angle in a specified range. Figure 3 shows the four derived nanoantennas for the four communication wavelengths, where the specified range for the oblique angles is from $-30^{{}^{\circ}}$ to $30^{{}^{\circ}}$ and the distributions of the real part of the magnetic field are plotted for the four oblique angles $0^{{}^{\circ}}$, $10^{{}^{\circ}}$, $20^{{}^{\circ}}$ and $30^{{}^{\circ}}$.

The spectra of the derived nanoantennas are plotted in Figure 4a as a function of the wavelength and oblique angle. From the spectra, we can confirm decreased dependence on the oblique angle at the communication wavelengths. The emitting performances of the derived nanoantennas are presented in the form of the polar diagrams in Figure 4b. The broadened main lobe width of the polar diagrams shows that this inverse design method can be used to decrease the oblique-angle dependence of the coupling nanoantenna at the cost of coupling efficiency λ.

With the exception of coupling the free space and MIM waveguide and controlling the directivity and emitting performance, the inversely designed nanoantennas can further be utilized to configure a nanostructure-based meta-interface in a subwavelength size for controlling the propagation of a light wave. As sketched in Figure 5a1, the meta-interface was constructed from an array of periodic units and it divides the free space into two parts. The light wave is refracted by this meta-interface based on the interaction between the free photonics and periodic unit. The periodic unit is composed of one MIM waveguide and two nanoantennas, which are localized at the inlet and outlet of the MIM waveguide. Figure 5a2 shows two neighboring periodic units, where the gap between the two neighboring nanoantennas should be wide enough to ensure the ignorance of their coupling. In these sketches, the solid red lines and dashed blue lines represent the light rays and iso-phase lines, respectively; $k_{i}$ is the wave vector of the coming light wave; $k_{o}$ is the wave vector of the refracted light wave; $\theta_{b}$ is the refractive angle; and λ is the wavelength of the light wave. The typical width of the gap is one quarter that of the incident wavelength. The lattice size of the periodic unit is determined by the wavelength and refractive angle as $\lambda /\sin\theta_{b}$. In such a structural unit, the left-side nanoantenna is used to receive the coming lightwave, enhance the coupling of the free space and MIM waveguide, and transfer the free photonics into the plasmon; the right-side nanoantenna is used to efficiently emit the propagating plasmon from the MIM into the free space, where the plasmon mode is then transferred into free photonics with a wave vector in the desired direction; and the MIM waveguide is used to propagate the surface plasmons and decouple the two nanoantennas that are localized at its inlet and outlet.

Based on the introduced method, two nanoantennas for the wavelength 1550 nm were inversely designed (as shown in Figure 5b1 and Figure 5b2, respectively) corresponding to the oblique angles $0^{{}^{\circ}}$ and $30^{{}^{\circ}}$, together with their polar diagrams. The nanoantenna with the oblique angle $0^{{}^{\circ}}$ was localized at the inlet of the MIM waveguide as the receiving nanoantenna. The nanoantenna with the oblique angle $30^{{}^{\circ}}$ was localized at the outlet of the MIM waveguide as the emitting nanoantenna. The length of the MIM waveguide was 250 nm, which was wide enough for the decoupling of the receiving and emitting nanoantennas. One structural unit of the meta-interface with the thickness of 750 nm, together with its polar diagram, was then derived (as shown in Figure 5b3) in order to achieve the refraction performance with $\theta_{b}$. The polar diagram of the derived structural unit was computed by localizing a magnetic dipole at the center of the MIM waveguide. Based on the reciprocity principle of electromagnetics [51], the refraction performance of the conjugated structural unit could be confirmed by the two main lobes of the polar diagram. The effective aperture of this structural unit was 2.43 μm, which was 78.2% of the lattice size. The corresponding transmission efficiency was 63.8%. Figure 5c1 shows the meta-interface composed of the periodic units in Figure 5b3, together with the magnetic field around the meta-interface. Figure 5c2 shows the energy flux density around the structural unit, where the white arrows represent the Poynting vectors. According to the magnetic field around the meta-interface and the distribution of the Poynting vectors, the refraction performance can be confirmed for the meta-interface that was composed of the structural unit conjugated with the inversely designed nanoantennas. The inversely designed meta-interface can be experimentally verified by the setup that is sketched in Appendix C.

## 4. Conclusions

This paper has developed a topology optimization-based inverse design method for nanoantennas that are localized at the interface between free space and the metal–insulator–metal waveguide, where the free photon is transferred into a surface plasmon. For the incident waves with communication wavelengths, the level of dependence on the incident wavelength and oblique angle can be decreased for the inversely designed nanoantennas at the cost of broadened directivity and coupling efficiency, by implementing a topology optimization procedure.

It has been demonstrated that the inversely designed nanoantennas can be used to configure a nanostructured interface in the subwavelength size in order to control the refraction of the incident wave. Inspired by the configuring operation of the derived nanoantennas, we have predicted that the developed inverse design method can be used to find structural elements which can be assembled in much the same way as LEGO blocks are in order to achieve a desired photonic performance.

The structures that were derived in this paper were used to demonstrate the potential of topology optimization as a means to manipulate and guide light by using designed plasmonic structures, which are important in the area of integrated photonics. We believe that numerous unit operations and applications could benefit from this approach, which will be our focus in the future.

## Figures and Tables

**Figure 1 nanomaterials-11-03219-f001:**
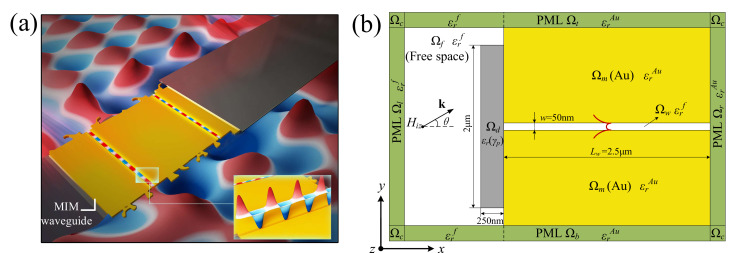
(**a**) The nanoantennas, which were inversely designed by using topology optimization, can couple the free space and MIM waveguide and efficiently excite the MIM plasmon modes at multiple wavelengths and under the oblique incidence. (**b**) Sketch of the 2D model of MIM waveguide. The model consists of a domain of free space $\Omega_{f}$
corresponding to the plane wave of the upper left part and the lower right part in Figure 1a, the sandwiched MIM waveguide composed of two Au domains $\Omega_{m}$, and a free space domain $\Omega_{w}$ corresponds to the middle gold part in Figure 1a. The design domain $\Omega_{d}$ for the nanoantennas is localized at the left endface of the MIM waveguide corresponds to both endfaces of the middle gold part in Figure 1a. The domain $\Omega =\Omega_{f}\cup \Omega_{d}\cup \Omega_{m}\cup \Omega_{w}$ is enclosed by perfectly matched layers (PMLs) $\Omega_{P}=\Omega_{t}\cup \Omega_{b}\cup \Omega_{l}\cup \Omega_{r}\cup \Omega_{c}$. The width and length of the MIM waveguide are the typical sizes of w=50 nm and $L_{w}=2.5\;\mu m$, respectively, which correspond to the part which filled with MIM plasmon modes in Figure 1a. These sizes can be fabricated by the processes introduced in [26,27]. The thickness and width of the design domain are 250 nm and 2 μm, respectively. $H_{iz}$ is the incident wave. θ is the oblique angle, which is the angle between the wave vector κ and positive x-direction.

**Figure 2 nanomaterials-11-03219-f002:**
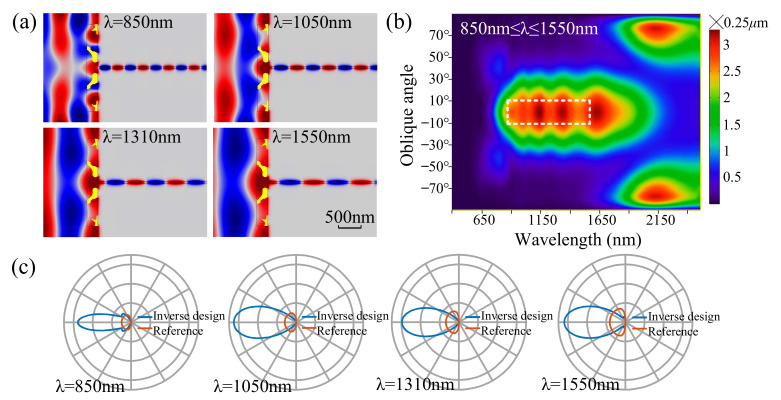
(**a**) Inversely designed MIM simultaneously working for the four communication wavelengths. (**b**) Effective-aperture spectrum of the inversely designed nanoantenna shown in Figure 2a. (**c**) Polar diagrams of the nanoantenna for the four communication wavelengths shown in Figure 2a.

**Figure 3 nanomaterials-11-03219-f003:**
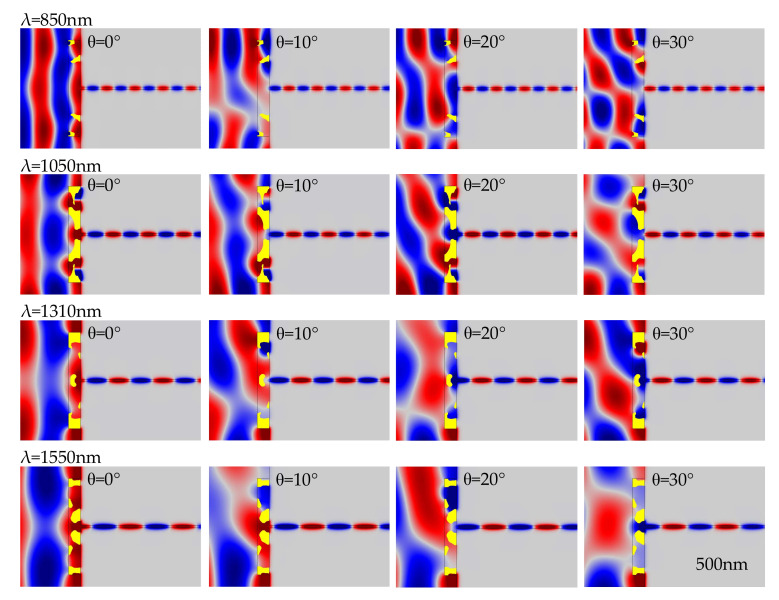
The four derived nanoantennas for the four communication wavelengths, where the specified range for the oblique angles is from $-30^{{}^{\circ}}$
to $30^{{}^{\circ}}$ and the distributions of the real part of the magnetic field are plotted for the four oblique angles $0^{{}^{\circ}}$, $10^{{}^{\circ}}$, $20^{{}^{\circ}}$ and $30^{{}^{\circ}}$.

**Figure 4 nanomaterials-11-03219-f004:**
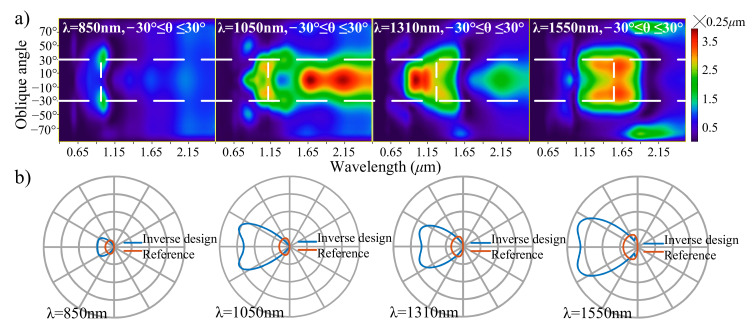
(**a**) Effective-aperture spectra as a function of the wavelength and oblique angle, for the four derived nanoantennas in Figure 3. (**b**) Polar diagrams of the four derived nanoantennas in Figure 3.

**Figure 5 nanomaterials-11-03219-f005:**
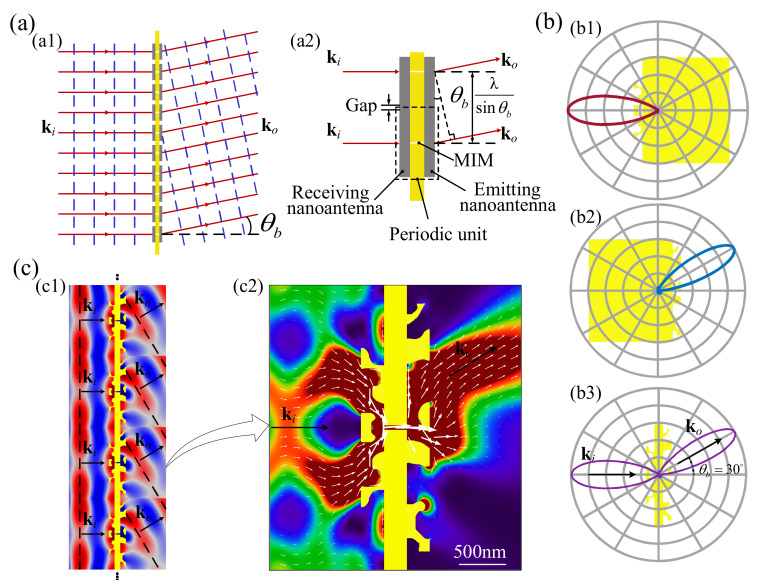
(**a**) Sketch of the meta-interface used to refract the propagation of light wave and two neighboring periodic units of the meta-interface. (**b**) Nanoantenna inversely designed and the conjugated structural unit of the meta-interface for the wavelength 1550 nm with the oblique angle $0^{{}^{\circ}}$, together with its normalized polar diagram. (**c**) Meta-interface composed of the structural unit shown in Figure 5b3, together with the magnetic field around the meta-interface and the energy flux density around the structural unit, where the white arrows represent the Poynting vectors.

## Data Availability

The data presented in this study are available on request from the corresponding author.

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
