# Peer review of "Nanoantennas Inversely Designed to Couple Free Space and a Metal–Insulator–Metal Waveguide"

_nanomaterials, 2021, doi:10.3390/nano11123219_

Round 1
Reviewer 1 Report
According to my opinion, the author has touched on a very important topic related to the light coupling to MIM waveguide. This kind of waveguide is widely used in designing plasmonic devices such as sensors which are numerically studied, however, there is no experimental demonstration yet due to the lack of light coupling capabilities. I am in favor of the publication of this paper; however, I have a few minor suggestions to be incorporated in the revised version.
- The author didn’t pay much attention to the editing of the paper template. Line 113 to 121 should be removed.
- Figure 2b, 4b, x-axis scale should be in nanometer (nm) to maintain the symmetry with the wavelength mentioned in the figures where H-field distribution is presented.
- What is the design domain as mentioned in line 79? Can the author explain a little bit more about it? What is the material of this element? And from where light is coupled to the inverse antenna?
- The design performance such as transmission efficiency should be mentioned in the Abstract and Conclusion section.
- The author has missed some important literature related to MIM waveguide sensing devices which have been recently published, consider citing them in the Introduction section. https://doi.org/10.3390/nano11102551, https://doi.org/10.1038/s41598-021-94143-2, https://doi.org/10.1038/s41598-021-98001-z.
- Is it possible for the author to draw/sketch an anticipated optical setup that can be used to couple the light into the MIM waveguide (by showing the optical elements which may be used for this technique)? It can raise the interest of the readers.
- Can the author comment, how the light can be collected from the MIM waveguide at the output port? Does it require some special geometry?
Reviewer 2 Report
The paper reports the inversely designed nanoantennas that can couple the free space and MIM waveguide and efficiently excite the MIM plasmon modes at multi wavelengths and under the oblique angles, by implementing the inverse design procedure using the topology optimization approach. I think, paper is interesting. I would propose some changes as follows:
- I would recommend approving the obtained results by comparing them with the experimental outputs.
- Authors are missing some recent articles in the field, such as
Analytic solution to field distribution in one-dimensional inhomogeneous media,
Analytic solution to field distribution in two-dimensional inhomogeneous waveguides, - I think, Authors should stress novelty of their work in comparison with others.
- Authors should mention origin of the absorptivity of the nanoantennas.
- Authors should indicate MIM waveguide in Fig. 1(a).
Round 2
Reviewer 2 Report
Accept.